# DNA methylation predicts age and provides insight into exceptional longevity of bats

Gerald S. Wilkinson [1,25✉], Danielle M. Adams[1], Amin Haghani[2], Ake T. Lu [2], Joseph Zoller [3], Charles E. Breeze [4], Bryan D. Arnold [5], Hope C. Ball[6], Gerald G. Carter[7], Lisa Noelle Cooper [6], Dina K. N. Dechmann [8,9,10], Paolo Devanna[11], Nicolas J. Fasel [12], Alexander V. Galazyuk[6], Linus Günther[13], Edward Hurme [1,9], Gareth Jones [14], Mirjam Knörnschild[10,13], Ella Z. Lattenkamp [11,15], Caesar Z. Li [3], Frieder Mayer[13], Josephine A. Reinhardt [16], Rodrigo A. Medellin [17], Martina Nagy [13], Brian Pope [18], Megan L. Power [19], Roger D. Ransome [14], Emma C. Teeling [19], Sonja C. Vernes [11,20,21], Daniel Zamora-Mejías [17], Joshua Zhang[2], Paul A. Faure [22], Lucas J. Greville [22], L. Gerardo Herrera M.[23], José J. Flores-Martínez[24] & Steve Horvath [2,3,25✉]

Exceptionally long-lived species, including many bats, rarely show overt signs of aging, making it difficult to determine why species differ in lifespan. Here, we use DNA methylation (DNAm) profiles from 712 known-age bats, representing 26 species, to identify epigenetic changes associated with age and longevity. We demonstrate that DNAm accurately predicts chronological age. Across species, longevity is negatively associated with the rate of DNAm change at age-associated sites. Furthermore, analysis of several bat genomes reveals that hypermethylated age- and longevity-associated sites are disproportionately located in promoter regions of key transcription factors (TF) and enriched for histone and chromatin features associated with transcriptional regulation. Predicted TF binding site motifs and enrichment analyses indicate that age-related methylation change is influenced by developmental processes, while longevity-related DNAm change is associated with innate immunity or tumorigenesis genes, suggesting that bat longevity results from augmented immune response and cancer suppression.

[1] Department of Biology, University of Maryland, College Park, MD, USA. [2] Department of Human Genetics, David Geffen School of Medicine, University of California, Los Angeles, CA, USA. [3] Department of Biostatistics, Fielding School of Public Health, University of California, Los Angeles, CA, USA. [4] Altius Institute for Biomedical Sciences, Seattle, WA, USA. [5] Department of Biology, Illinois College, Jacksonville, IL, USA. [6] Department of Anatomy and Neurobiology, Northeast Ohio Medical University, Rootstown, OH, USA. [7] Department of Evolution, Ecology and Organismal Biology, The Ohio State University, Columbus, OH, USA. [8] Department of Migration, Max Planck Institute of Animal Behavior, Radolfzell, Germany. [9] Department of Biology, University of Konstanz, Konstanz, Germany. [10] Smithsonian Tropical Research Institute, Panama, FL, USA. [11] Neurogenetics of Vocal Communication Group, Max Planck Institute for Psycholinguistics, Nijmegen, the Netherlands. [12] Department of Ecology and Evolution, University of Lausanne, Lausanne, Switzerland. [13] Museum für Naturkunde, Leibniz-Institute for Evolution and Biodiversity Science, Berlin, Germany. [14] School of Biological Sciences, University of Bristol, Bristol, UK. [15] Department Biology II, Ludwig Maximilians University Munich, München, Martinsried, Germany. [16] Department of Biology, State University of New York, Geneseo, NY, USA. [17] Instituto de Ecología, Universidad Nacional Autónoma de México, Ciudad Universitaria, Mexico City, Mexico. [18] Lubee Bat Conservancy, Gainesville, FL, USA. [19] School of Biology and Environmental Science, University College Dublin, Belfield, Dublin 4, Ireland. [20] Donders Institute for Brain, Cognition and Behaviour, Nijmegen, the Netherlands. [21] School of Biology, The University of St Andrews, Fife, UK. [22] Department of Psychology, Neuroscience and Behaviour, McMaster University, Hamilton, ON, Canada. [23] Estación de Biología de Chamela, Instituto de Biología, Universidad Nacional Autónoma de México, San Patricio, Mexico. [24] Departamento de Zoología, Instituto de Biología, Universidad Nacional Autónoma de México, Ciudad de México, Mexico. [25]These authors contributed equally: Gerald S. Wilkinson, Steve Horvath. ✉email: wilkinso@umd.edu; shorvath@mednet.a.ucla.edu

DNA methylation (DNAm) influences many processes including development[1], gene regulation[2], genomic imprinting[3], X chromosome inactivation[4], transposable element defense[5], and cancer[6]. Over 75% of cytosine-phosphate-guanine (i.e., CpG) sites are typically methylated in mammalian cells, but global DNAm declines with age, which can lead to loss of transcriptional control and either cause or contribute to, deleterious aging effects[7]. Conversely, DNAm often increases (i.e., shows hypermethylation) at CpG islands, which are CpG-dense regions often found in gene promoter regions near transcription start sites (TSS)[8–10]. Age-related changes in DNAm can be used to predict age in humans[11,12] and are beginning to be used to predict age in other species[13–17]. Given that the aging of wild animals typically requires long-term mark-recapture data or lethal tissue sampling, an accurate, noninvasive aging method would enable the study of age-associated changes in traits critical for survival, such as sensory perception, metabolic regulation, and immunity in a variety of long-lived species.

DNAm has also been used to predict lifespan in humans[18–20]. Intriguingly, interventions known to increase lifespan in some mammals, such as caloric restriction, reduce the rate at which methylation changes[21,22]. Moreover, a comparison of age-related changes in DNAm across species[22,23] suggests that DNAm rate also varies with lifespan. However, comparative studies have so far used different methods on a few primate, rodent, or canid species[22,23] making it difficult to determine reasons for methylation differences.

The distribution and function of genomic regions that exhibit age or longevity-related changes in DNAm are not fully understood[20,24]. In humans, hypermethylated age-associated CpG sites tend to be near genes predicted to be regulated by transcription factors involved in growth and development, whereas hypomethylated sites are near genes from more disparate pathways[25]. A recent study in dogs also found that age-related hypermethylated sites are near genes that influence developmental processes[14,17]. Human aging has been associated with modification of histone marks and relocalization of chromatin-modifying factors in a tissue-dependent manner[26]. Comparative analysis of CpG density in conserved gene promoter regions has revealed that CpG density is positively related to lifespan in mammals[27], as well as other vertebrates[28], but the genes involved were not enriched for any pathway or biological process.

Bats have great potential for providing insight into mechanisms that reduce deleterious aging effects because species from multiple independent lineages have maximum lifespans more than four times greater than similar-sized mammals[29] despite tolerating high viral loads[30,31] and showing few signs of aging. Here, we use a custom microarray that assays 37,492 conserved CpG sites to measure DNAm from known-aged individuals of 26 species of bats and address three questions. (1) How accurately can chronological age be estimated in bats? (2) Does an age-related change in DNAm predict maximum lifespan? (3) What genes are nearest the sites where DNAm changes as a function of age or differences in longevity between species? We find that DNAm can predict the age of individual bats with high accuracy. At the species level, the rate of change in DNAm at age-associated sites also predicts maximum lifespan. CpG sites that are informative for age or longevity are more likely to gain methylation and be near promoter regions of transcription factors involved in developmental processes. Longevity-associated sites are, in addition, enriched for genes involved in cancer suppression or immunity.

## Results

**Predicting individual age using DNAm.** DNAm profiles were analyzed from 712 wing tissue biopsies taken either from captive or free-ranging individuals of known age representing 26 species and six families of bats. Probe sequences were mapped in the genomes of nine of these species (Supplementary Table 1) and a total of 35,148 probes were located in at least one bat genome. For the 2340 probes not mapped in any bat, the median DNAm mean (0.496) and coefficient of variation (CV = 0.051%), were nearly identical to the 62 human SNP probes on each array (median mean = 0.500, median CV = 0.032%). In contrast, probes that mapped to at least one bat had DNAm means ranging from 0.006 to 0.995 with a median of 0.634. To predict age, we used sites mapped in one or more species from the taxonomic group of interest, i.e., order, genera or species.

Similar to human epigenetic clocks[12,20], elastic-net regression accurately predicted chronological age from a linear combination of DNAm beta values (henceforth DNAmAge) using 162 CpG sites. Leave-one-out (LOO) cross-validation shows that DNAm can predict age with a median absolute error (MAE) of 0.74 years (Fig. 1a). Limiting the analysis to smaller taxonomic groups (species or genera) can improve accuracy if sufficient data are available. For example, the correlation between chronological and DNAmAge in a LOO cross-validation analysis for 40–50 samples from a single species can be 0.96 or higher (Fig. 1b, c; Supplementary Fig. 1); a similar analysis on 176 samples from six *Pteropus* species gave a correlation of 0.97 (MAE = 0.77 years, Supplementary Fig. 2b). Thus, DNAm from a wing tissue sample for any of these species can reveal the animal's age at the time of sampling to within a year.

To assess how well DNAm might predict age in a species not represented by our samples, we conducted a second cross-validation analysis in which data for one species was left out (leave-one-species out; LOSO) and ages were predicted for that species using a clock estimated from the remaining 25 species. This analysis (Fig. 1d) resulted in a correlation between observed and predicted age of 0.84 (MAE = 1.41 years). The LOSO analysis also showed that DNAm consistently underestimates age in some species, while overestimating age in others. For example, *Desmodus rotundus* (sp. 5, Fig. 1d) samples exhibit lower values of DNAmAge (suggesting lower aging rates) than *Phyllostomus hastatus* (sp. 15, Fig. 1d) samples, consistent with the longer lifespan of *D. rotundus*[29].

**Predicting species longevity from DNAm.** To determine if the rate of DNAm change predicts variation in maximum lifespan among species, we incorporated a recent bat phylogeny[32] into a generalized least squares regression (PGLS) to predict the longevity quotient (LQ)—the ratio of observed to expected maximum lifespan for a mammal of the same body size[29]. We first identified a common set of age-associated CpG sites for this analysis by conducting a meta-analysis of all age-DNAm correlations by the probe for 19 bat species with 15 or more samples ("Methods"). The top 2000 age-associated sites (henceforth, age differentially methylated positions or age DMPs) consist of 1165 sites that show age-associated hypermethylation and 835 sites exhibiting age-associated hypomethylation. Both mean rates of hypermethylation and hypomethylation predict LQ, such that long-lived species have lower rates of DNAm change (Fig. 2a, b). A PGLS analysis on maximum lifespan with body mass as a covariate gave very similar results (Supplementary Table 2). Assuming that the rate of change in DNAm reflects epigenetic stability, these results suggest that better epigenetic maintenance is associated with a longer maximum lifespan, independent of body size, across bats.

**Identifying age and longevity-associated genes.** To identify DMPs associated with differences in LQ (henceforth longevity

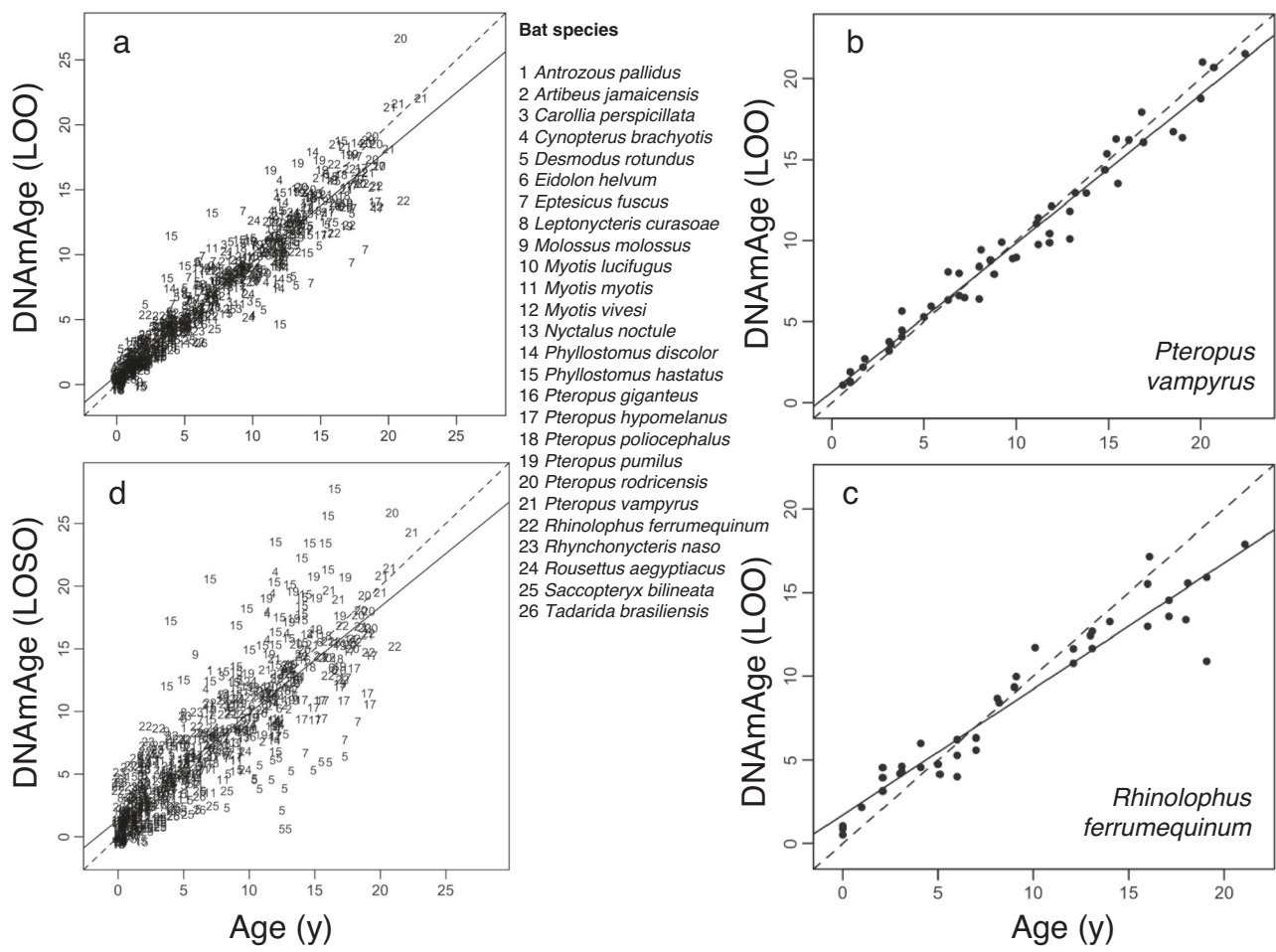

**Fig. 1 Epigenetic clocks accurately predict the chronological age of bats. a** Leave-one-out (LOO) cross-validation based on penalized regression gave a correlation of 0.95 with a median absolute error (MAE) of 0.74 years between observed and predicted (DNAmAge) age (after square-root transform) for 26 bat species. To ensure an unbiased cross-validation analysis, we allowed the number of CpGs to change with the respective training data. **b** LOO cross-validation based on penalized regression of 51 *Pteropus vampyrus* samples gave a correlation of 0.99 with MAE of 0.72 years between observed and predicted age. **c** LOO cross-validation based on penalized regression of 40 *Rhinolophus ferrumequinum* samples gave a correlation of 0.96 with MAE of 1.11 years between observed and predicted age. **d** Cross-validation analysis in which the DNAm data for one species was left out (LOSO) and ages are predicted for that species using a clock estimated with the remaining data. The resulting correlation between observed and predicted age is 0.84 (MAE = 1.41 years). Additional epigenetic clocks for individual species and genera are in Supplementary Figs. 1 and 2.

DMPs), we compared relationships between DNAm and age for three long-lived species and two short-lived species (cf. Fig. 2) from four bat families. Longevity DMPs have a significant (BY FDR = 0.05) interaction term between age and longevity type in a linear mixed model with species as a random effect (see Methods, Supplementary Fig. 3). We identified 1491 longevity DMPs, including 694 in which short-lived species gain DNAm faster with age and 797 in which short-lived species lose DNAm faster.

Age and longevity DMPs are widely distributed in the genome but differ in relative abundance across chromosomes (Fig. 3a, b). For example, of the 1077 probes that map to chromosome 1 (syntenic with the human X chromosome) in *R. ferrumequinum* (a long-lived bat with the most mapped probes, 30,724, Supplementary Table 1) only 12 are age-associated while 46 are longevity-associated. Not surprisingly, 596 of 753 sites (79.2%) that differ in DNAm values between the sexes across species are on the *R. ferrumequinum* X chromosome. Sex DMPs are independent of age DMPs (6.1% overlap, $P = 0.32$, Fisher's Exact Test, FET) and longevity DMPs (5.2% overlap, $P = 0.10$, FET). When limited to promoter regions, almost all age and longevity DMPs exhibit hypermethylation (Fig. 3c). Change in DNAm with respect to age correlates with a change in DNAm with respect to

longevity ($r = 0.454$, $P < 0.0001$), which results in significant overlap among longevity and age DMPs ($P < 0.0001$, FET, Fig. 3d and Supplementary Fig. 5a) and among unique genes near those DMPs (Fig. 3e and Supplementary Fig. 5b).

Even though about 7000 unique CpG probe sequences on the mammalian methylation array are unmapped in a bat genome (Supplementary Table 2), the mapped CpG sites are typically (median = 93%) nearest the same gene in any pair of bats (Supplementary Fig. 4c). Furthermore, genomic regions occupied by age and longevity DMPs are similar among bat species (Supplementary Fig. 4). For example, 68% of 2874 probes that map to a promoter region in the short lifespan species, *M. molossus*, also map to a promoter region in the distantly related long lifespan species, *R. ferrumequinum* (Fig. 4b). Promoter regions are enriched for hypermethylating, but not hypomethylating, age, and longevity DMPs in *M. molossus* (Fig. 4c, d) and other bats (Supplementary Fig. 5c, d). In bat genomes where CpG islands have been identified (e.g., *R. ferrumequinum*) hypermethylating age DMPs are much more likely than hypomethylating age DMPs to be located in CpG islands ($P < 0.0001$, FET); the same is true for longevity DMPs ($P < 0.0001$, FET).

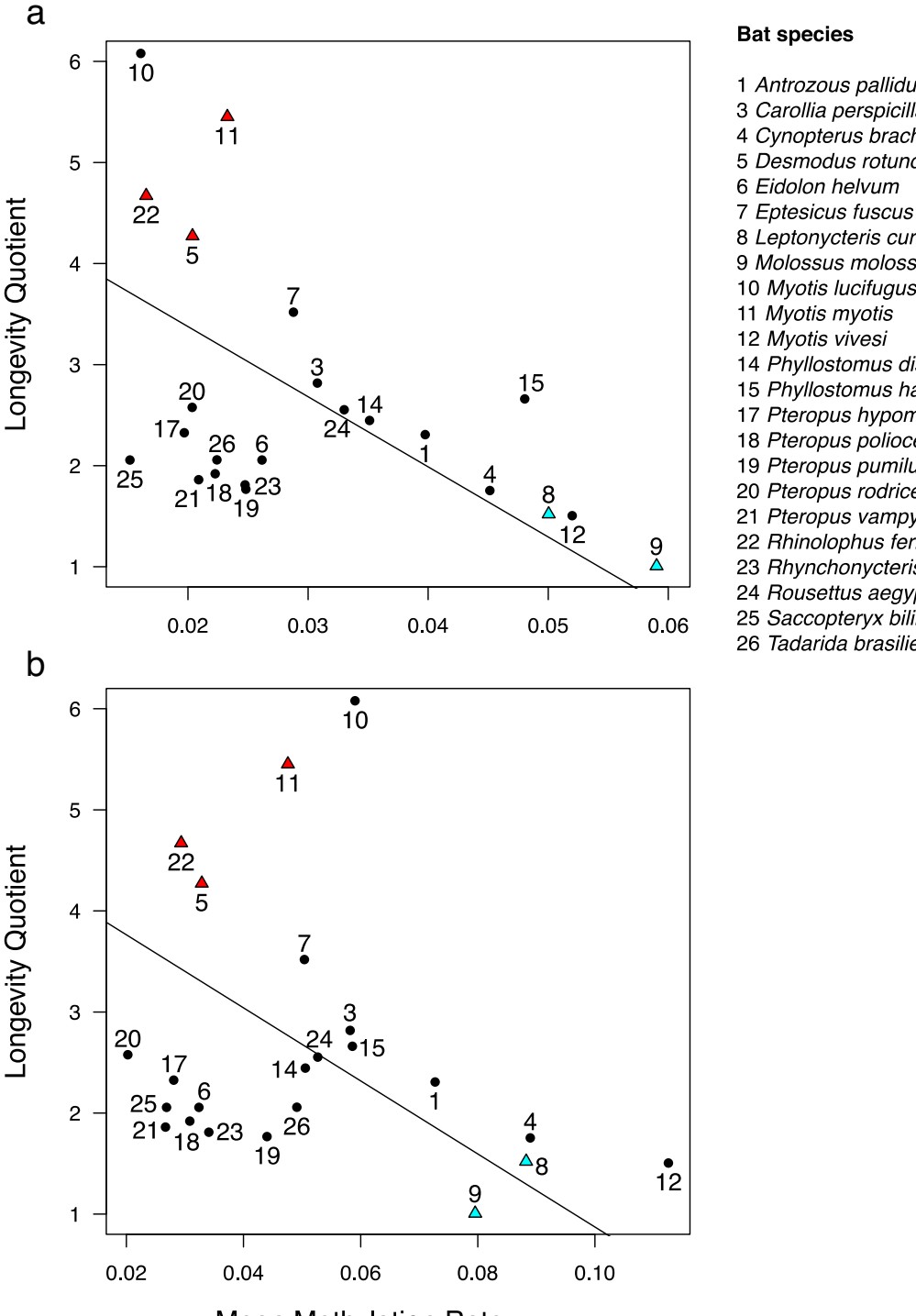

**Bat species**

1 *Antrozous pallidus*
3 *Carollia perspicillata*
4 *Cynopterus brachyotis*
5 *Desmodus rotundus*
6 *Eidolon helvum*
7 *Eptesicus fuscus*
8 *Leptonycteris curasoae*
9 *Molossus molossus*
10 *Myotis lucifugus*
11 *Myotis myotis*
12 *Myotis vivesi*
14 *Phyllostomus discolor*
15 *Phyllostomus hastatus*
17 *Pteropus hypomelanus*
18 *Pteropus poliocephalus*
19 *Pteropus pumilus*
20 *Pteropus rodricensis*
21 *Pteropus vampyrus*
22 *Rhinolophus ferrumequinum*
23 *Rhynchonycteris naso*
24 *Rousettus aegyptiacus*
25 *Saccopteryx bilineata*
26 *Tadarida brasiliensis*

**Fig. 2 Species longevity is predicted by the mean rate of DNAm change. a** After controlling for phylogeny using phylogenetic generalized least squares regression, mean DNAm rate at 1165 hypermethylating age DMPs correlate with longevity ($r = -0.704$, $t = -4.95$, $P = 6.79\text{e}{-5}$), **b** as does mean DNAm rate at 835 hypomethylating age DMPs ($r = -0.682$, $t = -4.27$, $P = 3.42\text{e}{-4}$). Species longevity is represented by the longevity quotient (LQ), which is the ratio of the observed species maximum lifespan to the maximum lifespan predicted for a nonflying placental mammal of the same body mass[29]. For example, the maximum longevity of *Myotis lucifugus* (10) is over six times longer than expected, while the maximum longevity of *Molossus molossus* (9) is equal to an average placental mammal of the same body size. The five species used for identifying longevity DMPs by the difference in methylation rate are indicated by red triangles (long-lived) and blue triangles (short-lived). Only species with more than ten samples are included.

Given that regions near promoters contain more age and longevity DMPs than expected, we evaluated the genes nearest those DMPs for possible functions. In view of the overlap in age and longevity DMPs, not surprisingly, genes with age or longevity DMPs in promoter regions show similar patterns of enrichment among biological process categories, i.e., developmental process, transcription, and regulation of transcription are enriched in *M. molossus* (Fig. 4e). Genes with age DMPs in

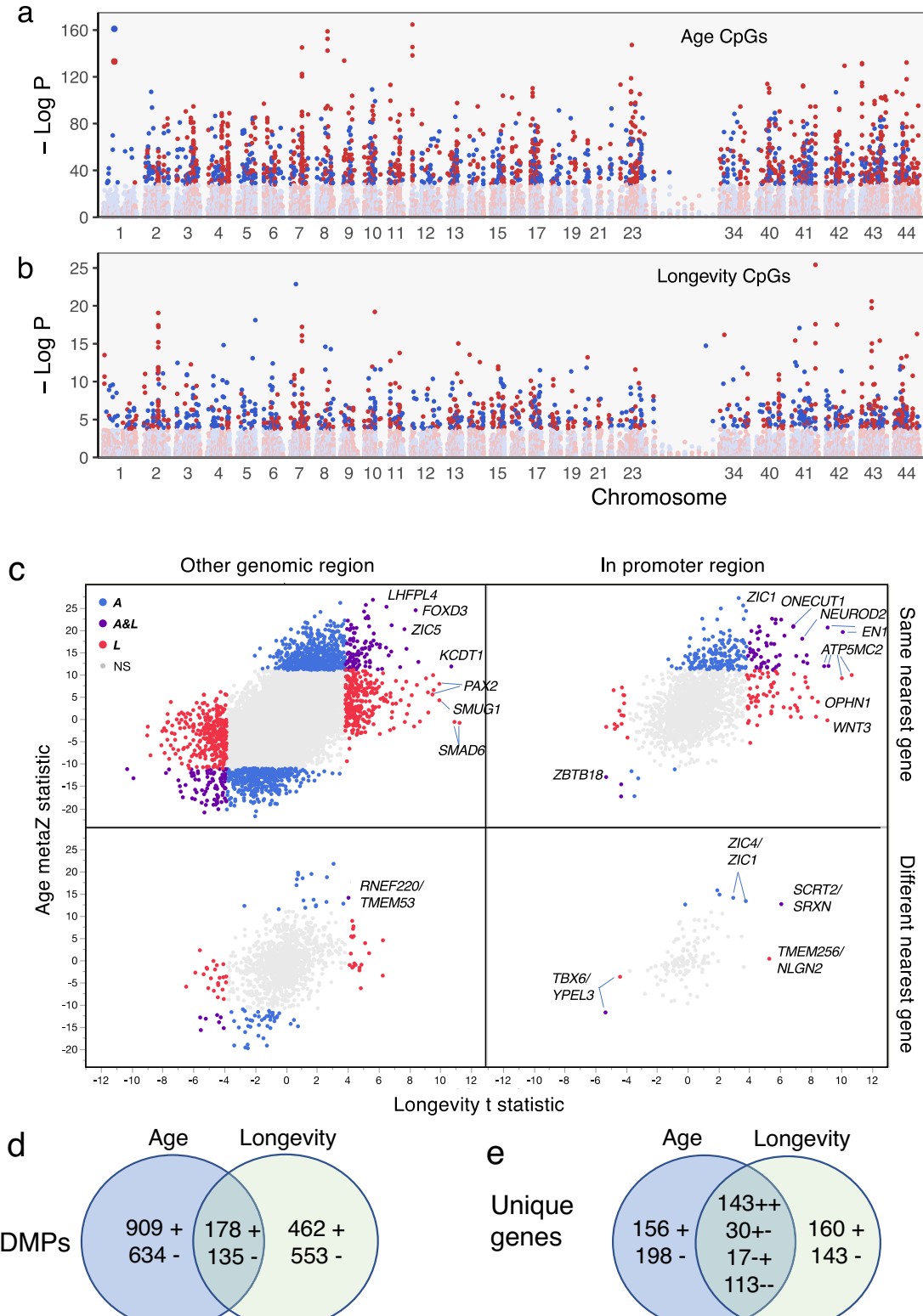

promoter regions are further enriched for multicellular organism development. With regard to protein class, gene lists for both age and longevity DMPs are enriched for homeodomain transcription factors containing helix-turn-helix motifs (Fig. 4f). These patterns are characteristic of other bat species too (Supplementary Fig. 6), although the gene list composition varies. For example, 142 hypermethylated age genes were identified across the four bat genomes used for identifying

longevity DMPs. Of these genes, 89 exhibited the same DMP-gene association in at least 3 of the 4 genomes.

Comparisons between the age and longevity-related genes and several relevant gene lists provide additional evidence for gene function. For example, hypermethylated age genes in bats strongly overlap hypermethylated age genes recently reported for dogs[17] (e.g., 83 of 143 hypermethylated dog genes are also related to age in the short lifespan *M. molossus*,

**Fig. 3 Differentially methylated positions (DMPs) for age and longevity are widely distributed and partially overlap. a** Negative log *P* for age-associated DMPs plotted against location on each *Rhinolophus ferrumequinum* chromosome. The top 2000 age-associated DMPs are darkened with increasing DNAm indicated by red and decreasing DNAm indicated by blue. Hypermethylated age-DMPs are underrepresented on chromosome 1, which is syntenic with the human X chromosome. **b** Longevity DMPs are also distributed across all *R. ferrumequinum* chromosomes. Darkened symbols indicate 1491 significant (BY 5% FDR) longevity DMPs with colors indicating DNAm direction as in (**a**). **c** Effect of DNAm change on age plotted against the effect of DNAm change on longevity (see "Methods") illustrates the association between age and longevity effects. Significant sites are colored blue for age, red for longevity, and purple for both age and longevity. Symbols for the orthologous gene with the nearest transcription start site (TSS) to the DMP are indicated for a sample of extreme age and longevity DMPs. Bottom panels indicate DMPs that map to different genes in the short-lived species, *M. molossus*, and the long-lived species, *R. ferrumequinum*, with the *M. molossus* gene indicated after /. Note that most extreme age and longevity DMPs in promoter regions (i.e., −10,000 to +1000 bp from the TSS) are in the upper right panel, i.e., nearest the same gene in both species. **d** Age DMPs overlap 17% with hypermethylating (+) and hypomethylating (−) longevity DMPs in *M. molossus*. Long-lived bat species show similar patterns (Supplementary Fig. 5a). **e** Number of unique genes nearest age and longevity DMPs for *M. molossus*. Signs on numbers in the overlap region indicate methylation direction for age then longevity. Long-lived bat species show similar patterns (Supplementary Fig. 5b).

$P = 4.57e−54$, FET). In contrast, only 5 of 60 hypomethylated dog genes are related to age in *M. molossus* ($P = 0.21$, FET). *Molossus molossus* age genes are not enriched for immunity genes ($P = 0.24$, FET) or genes that frequently mutated in cancer ($P = 0.21$, FET). However, *M. molossus* longevity genes exhibit significant overlap with genes involved in immunity ($P = 0.002$, FET) and genes frequently mutated in human tumors ($P = 0.016$, FET, Fig. 4g). Similar overlap patterns among immunity, longevity, and tumor-mutated genes also exist for long-lived bats (Supplementary Fig. 6).

While methylation in a promoter region can influence transcription, transcription regulation can also result from interactions among DNA-bound proteins that are in proximity due to chromatin folding[33]. To evaluate the possibility of either short or long-range transcriptional regulation, we used eFORGEv.2.0[34] to predict how DMPs likely influence regulatory regions. This program first identifies probe sequences as being associated with five core histone marks or 15 predicted chromatin states in prior epigenomic studies using over 100 cell lines from multiple tissue sources, then uses permutation tests against the species genomic background to determine which histone marks or chromatin states occur nonrandomly. Using probes mapped in the long lifespan species *Desmodus rotundus* as background, we find that age and longevity DMPs exhibiting hypermethylation are enriched for repressive histone H3 trimethylated at lysine27 (H3K27me3) and active H3K4me1 marks in relevant cell lines (Fig. 5a, b). Hypomethylated age DMPs are enriched in all tissues for H3K9me3, while hypomethylated longevity DMPs show no enrichment (Fig. 5a, b). Analysis of predicted chromatin states reveals that hypermethylated age DMPs are enriched in all tissues for repressed polycomb complexes, while hypomethylated age DMPs are enriched for quiescent chromatin states (Fig. 5c). Longevity DMPs, both hypermethylating and hypomethylating, also show enrichment for quiescent states, as well as enrichment for repressive polycomb complexes or enhanced bivalent states in some tissues (Fig. 5d).

Transcription factor (TF) motifs identified in DMP probe sequences that are involved in cell cycle regulation and genome stability are enriched among hypermethylating age sites (Fig. 5e). Several of those transcription factors, including Cut-like homeobox 1 (CUX1), AT-rich interaction domain 3A (ARID3), and E2f transcription factor 1 (E2F) are involved in cell cycle regulation[35–37], while others, such as Zinc finger protein 161 (ZFP161), are involved in genome stability[38]. In contrast, hypomethylating age sites only overlap three TF clusters, one of which, IRF7, is a master regulator of the interferon-dependent innate immune response in bats[39].

Longevity TF motifs are largely independent of age TF motifs (Fig. 5e), with one exception, c203-AP2/2, a cluster including Transcription factor AP-2 gamma (TFAP2C), which is involved

in epidermal cell lineage commitment[40] and regulation of tumor progression[41]. The other longevity TF motifs also have known associations with tumorigenesis. The c221-GCM1/3 transcription factor cluster includes Pleiomorphic adenoma gene-like 1 (PLAGL1), a protein that suppresses cell growth. The gene that encodes this protein is often methylated and silenced in cancer cells[42]. CNOT3 acts as a tumor suppressor in T-cell acute lymphoblastic leukemia (T-ALL)[43] but can also facilitate the development of non-small cell lung cancer[44]. Finally, HIC1, Hypermethylated in cancer 1 protein, acts as a tumor suppressor and is involved in the regulation of p53 DNA damage responses[45]. Only a single TF motif, HD/5 in the BARHL2 group[46], was associated with hypomethylated longevity DMPs.

Enrichment analyses[47] using the age and longevity gene lists for *M. molossus* identify several key gene regulators that are significantly associated with hypermethylated sites, but none with hypomethylated sites (Fig. 5f and Supplementary Fig. 6c). Orthodenticle homeobox 2 (OTX2) and Re1 silencing transcription factor (REST) are associated with both age and longevity, whereas other predicted TFs largely differ between age and longevity. REST is induced during human aging and represses neuronal genes that promote cell death[48]. Note that four of nine transcription regulators predicted to be associated with longevity frequently undergo mutations in human tumors and three are involved in innate immunity (Fig. 5f).

## Discussion

As with other species[13,14,17,49], age-related changes in DNAm occur throughout bat genomes. While 162 CpG sites are sufficient to predict chronological age, these represent only a small fraction of the sites that correlate with age, because penalized regression excludes highly correlated variables to avoid multi-collinearity. Consequently, we carried out a meta-analysis that correlated methylation at individual CpG sites with age across species to identify age DMPs. At these sites, long-lived species exhibit a lower rate of change in DNAm, while short-lived species exhibit faster increases in DNAm. How those changes contribute to longevity is not entirely clear, but our results suggest several key transcriptional regulators are involved and modulate the rate at which DNAm changes between short and long-lived species.

Our results are consistent with an epigenetic clock theory of aging that connects beneficial developmental and cell maintenance processes to detrimental processes causing tissue dysfunction[20]. A large body of evidence links age-related hypermethylated sites to genes and genomic regions that influence developmental processes[9,10,17]. We find that the sites that gain DNAm with age also tend to be in CpG islands, consistent with studies in humans[50]. In contrast, we find little enrichment for genes associated with hypomethylated sites, and these genes are

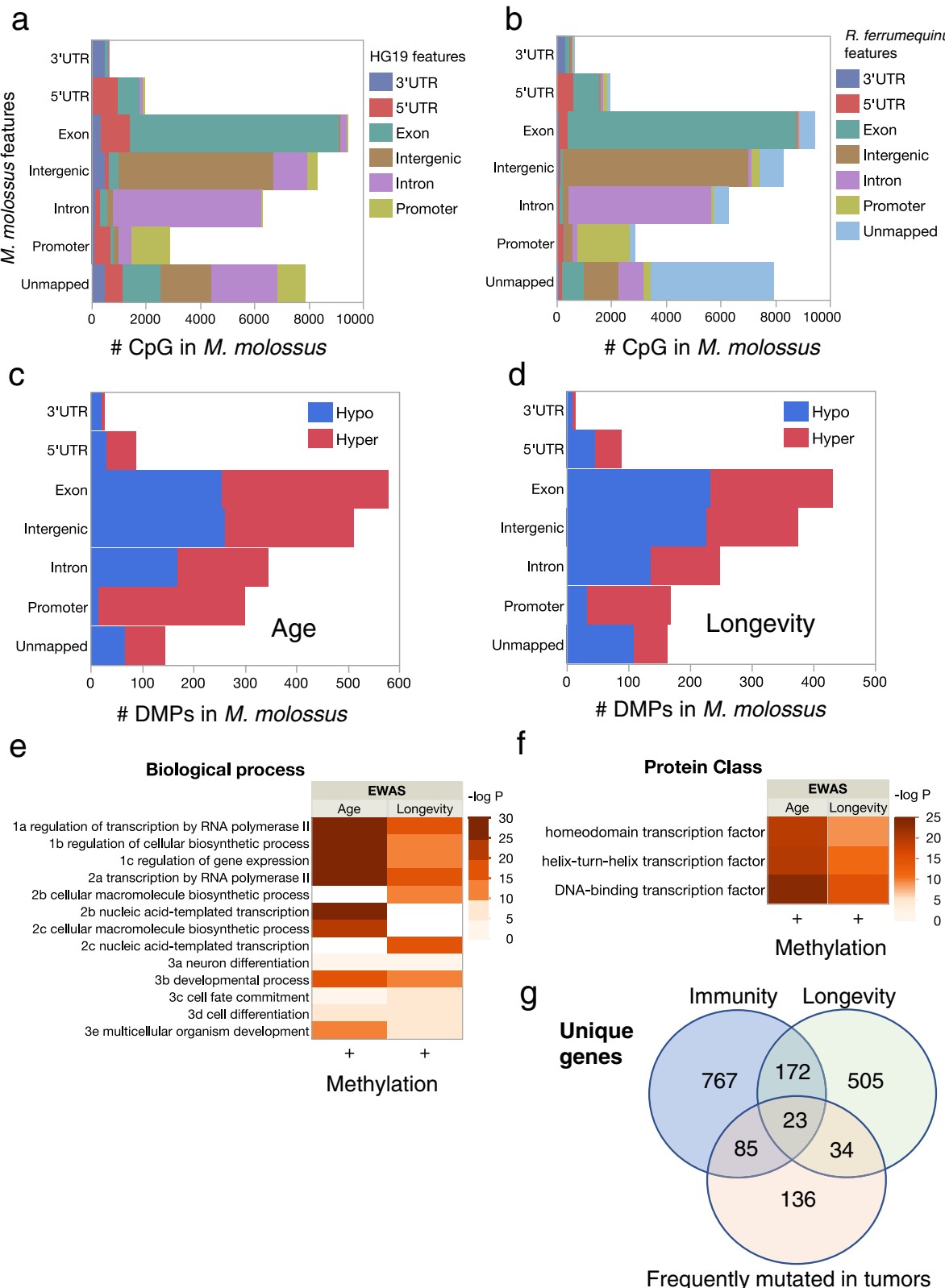

less likely to be shared across species. We interpret these results to indicate that DNAm loss with age is widespread and not concentrated in particular pathways. DNAm gain with age, on the other hand, occurs predictably near genes involved in many of the same developmental processes in humans, mice, dogs, and bats, consistent with a shared mammalian origin.

Our analyses are based entirely upon wing biopsy samples and the reported DNAm patterns could differ by tissue, as has been frequently observed[8]. However, bat wing tissue is capable of unusually rapid regeneration[51] and consists of multiple tissue types[52], making it particularly useful for measuring age-related changes in DNAm. In addition, these non-lethal biopsies are

**Fig. 4 Age and longevity DMPs are enriched in promoter regions of genes associated with immunity and cancer. a** CpG annotation for the short-lived bat, *M. molossus*, in comparison to genome regions where probes map to the human genome (HG19) shows that fewer than half of the probes that map to a promoter region in the bat also map to a promoter region in human (see also Supplementary Fig. 4). Colors indicate genomic regions in the human genome as indicated in the legend. **b** In contrast, CpG annotation comparison between two phylogenetically distant bat species, *M. molossus*, and *R. ferrumequinum*, indicates greater probe conservation with respect to gene proximity (see also Supplementary Fig. 4). Colors indicate genomic regions in the genome of *R. ferrumequinum*. **c** The top 2000 age DMPs are highly enriched near promoter regions with over 95% exhibiting hypermethylation in *M. molossus* and other bats (Supplementary Fig. 5). Red indicates DMPs associated with increasing DNAm, blue indicates DMPs associated with decreasing DNAm. **d** The 1491 longevity DMPs are also enriched in promoter regions with over 80% exhibiting hypermethylation in *M. molossus* and other bats (Supplementary Fig. 5). Color as in (**d**). **e** Enriched biological processes for unique *M. molossus* genes from promoter regions are only significant for hypermethylating age and longevity DMPs. Only three significant GO terms from each parent–child group are shown to minimize redundancy. **f** Enrichment analysis of protein class for unique *M. molossus* genes from promoter regions reveals significant enrichment of helix-turn-helix transcription factors (TF) only for hypermethylated DMPs associated with age and longevity. Cell color indicates significance (negative log *P* for GO terms with adj*P* < 10e−4) of enrichment in (**e**) and (**f**). **g** Overlap between genes associated with longevity, innate immunity, or frequently mutated in human tumors identified in *M. molossus*. Enrichment analyses using genome annotations from other bat species produce similar results (Supplementary Fig. 6).

---

relatively easy to obtain from wild-caught bats, thus allowing for future longitudinal and cross-sectional studies of epigenetic aging.

DNAm of genes suppressed in stem cells is a hallmark of cancer[10]. Several lines of evidence suggest that bat genes with longevity DMPs are important for cancer suppression and provide enhanced immunity. First, these genes disproportionately include many known to mutate frequently in human cancers or involved in innate immunity. Second, several transcription factors identified by motif analysis act as tumor suppressors, such that if they are silenced by methylation in older individuals, tumor formation should be more likely. Third, among the transcription factors identified from the list of genes with hypermethylated sites in promoter regions, several of them mutate in human cancers. While bats are not immune from cancer[53], genetic adaptations for tumor suppression have been described for *Myotis brandtii*[54] and *M. myotis*[55] to help explain the extreme longevity of those species. Bats also have genetic mechanisms that enable strong antiviral immune responses without inducing damaging inflammatory reactions that may enable them to tolerate high levels of viral exposure[30,31,56]. The results of this study are consistent with the hypothesis that enhanced epigenetic stability, especially associated with innate immunity and cancer suppression genes, facilitates exceptional longevity in bats.

## Methods

**Wing tissue samples**. Wing punches were taken from 778 individually marked animals that were either kept in captivity (15 species) or recaptured as part of long-term field studies (11 species). We excluded 42 samples because we did not have independent evidence to confirm minimum age estimates. For 630 samples the individual was marked shortly after birth, so age estimates were exact. For the remainder, age represented a minimum estimate because the individual was not initially banded as a juvenile. We used minimum age estimates when other evidence, such as tooth wear or time since initial capture, indicated that the minimum age estimate was likely to be close to the real age. In the Supplementary Methods, we provide additional information on when and where samples were taken from either captive or free-ranging animals and details of research permits and animal use protocols. We affirm that we have complied with relevant ethical regulations. The study was approved by the University of Maryland Institutional Animal Care and Use Committee (FR-APR-18-16).

After extraction DNA concentration was estimated with a QuBit and samples were concentrated, if necessary, to reach a minimum of 10 ng/μl in 20 μl. To estimate rates of methylation we limit analyses to the 23 species for which we had more than 10 samples from known-aged individuals. The maximum lifespan for each species was obtained from[29] or from captivity records and is listed along with the range of ages of individuals sampled in Table 1.

**DNAm profiling**. All methylation data were generated using a custom Illumina methylation array (HorvathMammalMethylChip40) based on 37,492 CpG sites[57]. Out of these 37,492 sites, 1951 CpGs were selected based on their utility for estimating human age in prior human biomarker studies. The remaining 35,541 probes were chosen due to their location in highly conserved 50 bp sequences with a terminal CpG site. The particular subset of species for each probe is provided in the chip manifest file at the NCBI Gene Expression Omnibus (GEO) platform

(GPL28271). Five bat genomes, *Pteropus vampyrus*, *P. alecto*, *Eptesicus fuscus*, *Myotis davidii* and *M. lucifugus*, were used in the design of the array.

Bisulfite conversion of DNA samples using the Zymo EZ DNA Methylation Kit (ZymoResearch, Orange, CA, USA), as well as subsequent Cy3 and Cy5 labeling, hybridization, and scanning (iScan, Illumina), were performed according to the manufacturers' protocols by applying standard settings. DNAm levels ($\beta$ values) were determined by calculating the ratio of intensities between methylated (signal A) and unmethylated (signal B) sites. Specifically, the $\beta$ value was calculated from the intensity of the methylated (M corresponding to signal A) and unmethylated (U corresponding to signal B) sites, as the ratio of fluorescent signals $\beta = $ Max $(M,0)/[$Max$(M,0) + $Max$(U,0) + 100]$. Thus, $\beta$ values range from 0 (completely unmethylated) to 1 (completely methylated). The SeSaMe method[58] was used to normalize $\beta$ values for each probe. A cluster analysis by species identified 24 samples as outliers, likely due to their low DNA concentrations. After excluding these, along with the 42 excluded due to insufficient age information, we retained 712 of the 778 samples for further analysis.

**Probe mapping and annotation**. We used sequences and annotations for ten bat genomes (Supplementary Table 1), which include six recently published reference assemblies[19], to locate each 50 bp probe on the array. The alignment was done using the QuasR package[59] with the assumption for bisulfite conversion treatment of the genomic DNA. For each species' genome sequence, QuasR creates an in-silico-bisulfite-treated version of the genome. The set of nucleotide sequences of the designed probes, which includes degenerate base positions due to the bisulfite conversion, was expanded into a larger set of nucleotide sequences representing every possible combination of degenerate bases. We then ran QuasR (a wrapper for Bowtie2) with parameters −k 2—strata—best −v 3 and bisulfite = undir to align the enlarged set of probe sequences to each prepared genome. From these files, we collected only alignments where the entire length of the probe perfectly matched the genome sequence (i.e., the CIGAR string 50 M and flag XM = 0).

Following the alignment, the CpGs were annotated based on the distance to the closest transcriptional start site using the ChIPseeker package[60]. A gff file with these was created using these positions, sorted by scaffold and position, and compared to the location of each probe in BAM format using Samtools. We report probes whose variants only mapped to one unique locus in a particular genome. Gene annotations for the ten bat genomes are available at http://hdl.handle.net/1903/26373.

The genomic location of each CpG was categorized as intergenic, 3′ UTR, 5′ UTR, promoter region (minus 10 kb to plus 1000 bp from the nearest TSS), exon, or intron. We identified X-linked probes in bat genomes by comparison to probes mapped to the X for the human genome, HG19. Tests for enrichment among genomic categories were performed with contingency or Fisher's Exact tests (FET) in JMP Pro v14.1 for the four species used to identify longevity-associated sites, i.e., one short-lived bat, *Molossus molossus*, and three long-lived bats, *Myotis myotis*, *Desmodus rotundus* and *Rhinolophus ferrumequinum*, representing four different bat families. We did not include *Leptonycteris yerbabuenae* in these analyses because no genome is available for that species. While most sites map to the same nearest gene, some differences exist. In the text, we present enrichment results for the short-lived species, *M. molossus*, but provide parallel results in Supplementary Figures for one or more of the long-lived species, *R. ferrumequinum*, *D. rotundus* and *M. myotis*.

**Creation of epigenetic clocks using penalized regression**. We developed epigenetic clocks for bat wing tissue by regressing chronological age on all CpGs that map to at least one of the ten bat genomes. To improve linear fit we transformed chronological age to sqrt(age + 1). Penalized regression models were created in the R package glmnet[61]. We investigated models produced by elastic net regression (alpha = 0.5). The optimal penalty parameters in all cases were determined automatically by using a tenfold internal cross-validation (cv.glmnet) on the training

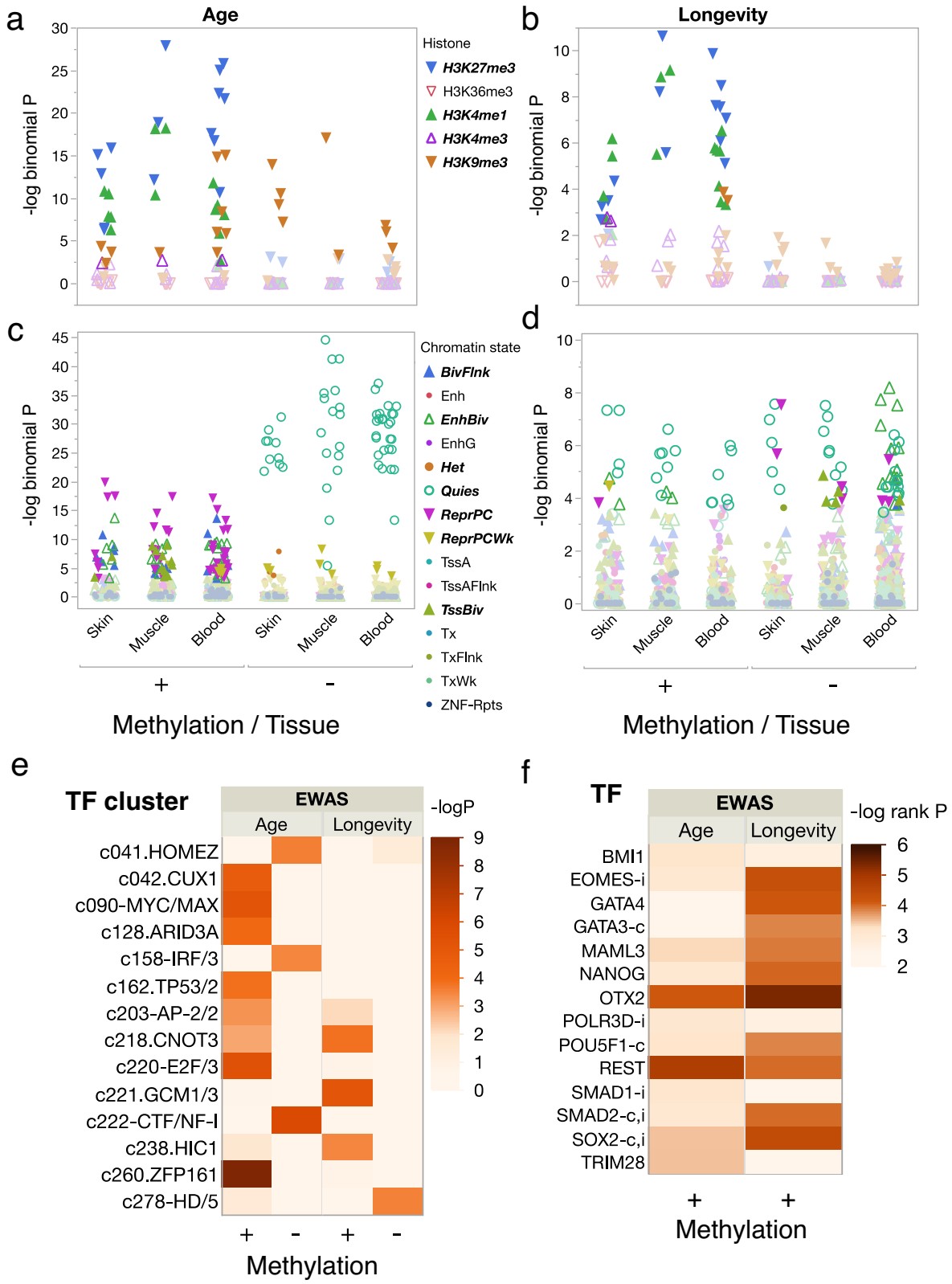

set. By definition, the alpha value for the elastic net regression was set to 0.5 (midpoint between Ridge and Lasso-type regression) and was not optimized for model performance. We performed two cross-validation schemes for arriving at unbiased estimates of the accuracy of the different DNAm based age estimators. One type consisted of leaving out a single sample (LOO) from the regression, predicting an age for that sample by regressing an elastic net on the methylation profiles of all other samples and iterating over all samples. We conducted LOO analyses using all samples from all species, using all samples from each species and

using all samples from several species in the same genus. The second type consisted of leaving out a single species (LOSO) from the regression, thereby predicting the age of each sample using the data for all other species.

**Differentially methylated positions (DMPs) for age and longevity**. To identify DMPs associated with age, we used WGCNA and METAL to compute the Pearson correlation coefficient between methylation level (β) and chronological age for each

**Fig. 5 Functional overlap analysis of DMPs reveals the role of key transcriptional regulators.** Histone marks (e.g., H3K27me3 = trimethylation of lysine 27 on histone H3) are denoted by shape and color as indicated in the legend for cell lines derived from skin, muscle, or blood for DMPs mapped in *Desmodus rotundus*, a long lifespan species, for (**a**) age and (**b**) longevity with darkened symbols indicating significance (BY 5% FDR) and ±indicating positive/negative rates of DNAm change. Enriched chromatin states (e.g., ReprPC = Repressed Polycomb) as predicted by a hidden Markov model for cell lines derived from skin, muscle, or blood are denoted by shape and color as indicated in the legend for (**c**) age and (**d**) longevity DMPs mapped in *Desmodus rotundus* with darkened symbols indicating significance (BY 5% FDR) and ±indicating positive/negative rates of DNAm change. **e** Transcription factor clusters enriched for hypermethylated (+) and hypomethylated (−) age or longevity DMPs with cell color indicating significance (negative log *P*, adj*P* < 10e −4) of overlap with predicted transcription factor binding sites in probe sequences using a hypergeometric test. **f** Top-ranked transcription factors associated with the change in expression of genes containing age or longevity DMPs in promoter regions in *M. molossus*, with integrative rank significance (see "Methods") indicated as negative log *P*. Genes frequently mutated in human tumors are indicated by (*c*), and those involved in innate immunity by *i*. Only genes with hypermethylated sites in promoter regions showed evidence of enrichment. Analyses using genome annotations from other bat species produce similar results (Supplementary Fig. 6).

**Table 1 Summary[a] of samples used for DNAm profiling.**

| Genus species (Family) | Source | #F | #M | Exact N | Yg | Old | Max age |
|---|---|---|---|---|---|---|---|
| *Antrozous pallidus* (V) | F | 21 | 2 | 1 | 0.2 | 7 | 14.8 |
| *Artibeus jamaicensis* (Ph) | C | 3 | 0 | 2 | 11 | 13 | 19.2 |
| *Carollia perspicillata* (Ph) | C | 17 | 15 | 32 | 0.2 | 10.5 | 17.0 |
| *Cynopterus brachyotis* (Pt) | C | 6 | 4 | 10 | 6.7 | 12.9 | 13.0 |
| *Desmodus rotundus* (Ph) | C | 27 | 17 | 41 | 0.3 | 17.3 | 29.9 |
| *Eidolon helvum* (Pt) | C | 17 | 7 | 24 | 3.4 | 16.5 | 21.8 |
| *Eptesicus fuscus* (V) | C | 18 | 41 | 59 | 0.3 | 18.3 | 23.0 |
| *Leptonycteris yerbabuenae* (Ph) | F | 5 | 6 | 7 | 0.2 | 5 | 10.1 |
| *Molossus molossus* (M) | F | 9 | 5 | 6 | 0.3 | 5.9 | 5.9 |
| *Myotis lucifugus* (V) | F | 11 | 0 | 1 | 0.1 | 5 | 34.0 |
| *Myotis myotis* (V) | F | 36 | 2 | 33 | 1 | 9 | 37.1 |
| *Myotis vivesi* (V) | F | 11 | 6 | 4 | 0.1 | 2 | 10.0 |
| *Nyctalus noctula* (V) | F | 3 | 0 | 2 | 0.9 | 2 | 12.0 |
| *Phyllostomus discolor* (Ph) | C | 31 | 19 | 42 | 0.1 | 17.7 | 18.0 |
| *Phyllostomus hastatus* (Ph) | F | 61 | 10 | 52 | 0.1 | 16.5 | 22.0 |
| *Pteropus giganteus* (Pt) | C | 0 | 4 | 4 | 10.9 | 14.2 | 44.0 |
| *Pteropus hypomelanus* (Pt) | C | 28 | 12 | 40 | 0.4 | 19.3 | 26.5 |
| *Pteropus poliocephalus* (Pt) | C | 10 | 6 | 16 | 6.1 | 16.7 | 23.6 |
| *Pteropus pumilus* (Pt) | C | 24 | 22 | 45 | 0.8 | 17.3 | 17.3 |
| *Pteropus rodricensis* (Pt) | C | 12 | 7 | 19 | 4 | 20.9 | 28.0 |
| *Pteropus vampyrus* (Pt) | C | 27 | 24 | 51 | 0.6 | 22.4 | 24.0 |
| *Rhinolophus ferrumequinum* (R) | F | 40 | 0 | 39 | 0.1 | 21.1 | 30.5 |
| *Rhynchonycteris naso* (E) | F | 6 | 16 | 15 | 0.1 | 6 | 8.5 |
| *Rousettus aegyptiacus* (Pt) | C | 8 | 8 | 3 | 5 | 14 | 22.9 |
| *Saccopteryx bilineata* (E) | F | 20 | 9 | 24 | 0.2 | 8.3 | 11.0 |
| *Tadarida brasiliensis* (M) | C | 9 | 10 | 15 | 0.2 | 6.2 | 12.0 |

[a]Family: E = Emballonuridae, M = Molossidae, Ph = Phyllostomidae, Pt = Pteropodidae, R = Rhinolophidae, V = Vespertilionidae; Source: F = field, C = captivity; #F, #M: number of samples for each sex; Exact N: number of individuals with exact age estimates; Yg, Old: youngest (Yg) and oldest (Old) individual sampled in years; Max age: maximum recorded age in years.

of the 37,492 sites for the 19 species with 15 or more samples (Table 1). The significance of each site across species was then evaluated using Stouffer's unweighted *z*-test[62]. CpG sites were ranked by significance and the top 2000 sites based on the correlation with untransformed age were selected for subsequent analyses and are referred to as age DMPs. For probes with contrasting patterns in different species, methylation direction was assigned based on the most frequent direction across species to ensure mean methylation rates are comprised of the same set of sites in each species. Because we used all sites on the array, some sites do not map to a unique position in one or more bat genomes. Supplementary Table 1 indicates how many sites map to each species.

To identify DMPs associated with longevity we compared methylation rates between three long-lived species (*R. ferrumequinum*, *D. rotundus*, and *M. myotis*) and two short-lived species (*M. molossus* and *L. yerbabuenae*). We chose these five species because they represent three independent lineages of increased longevity[29] and because high-quality genome assemblies are available for four of them[63]. We used a linear mixed-effects model (nlme) to fit methylation level (β) as a function of transformed chronological age (sqrt(age + 1)), longevity category, and their interaction, with species included as a random effect. We defined probes as longevity-associated if the p-value of the interaction term was less than 0.05 after Benjamini–Yekutieli (BY) false discovery rate (FDR) correction[64]. In this analysis, a positive interaction means a steeper positive slope for the short-lived species relative to the long-lived species. If the main effect of age is positive (hypermethylation) and the interaction is positive, then short-lived species are gaining methylation faster. If the main effect is

negative and the interaction is negative, then short-lived species are losing methylation faster.

**Phylogenetic analysis of bat longevity.** Using phylogenetic generalized least squares regression (PGLS) we tested the effect of the mean rate of methylation change on longevity using both the LQ and maximum longevity (log-transformed). LQ is the ratio of the observed species maximum lifespan to the maximum lifespan predicted for a nonflying placental mammal of the same body mass[29]. We present results for LQ in the text and for a model containing log(maximum longevity) and log(mass) in Supplementary Table 2. For each species with at least ten known-age samples, we calculated the mean rates of hypermethylation and hypomethylation using the top 2000 age-associated DMPs as described above. Hypermethylation and hypomethylation rates were tested separately. Phylogenetic relationships among bats are based on a recent maximum likelihood tree[32]. Models were fit via maximum likelihood using the gls function of the nlme R package and assume a Brownian model of trait evolution.

**Probe and gene enrichment analyses.** To determine how changes in methylation influence age and longevity, we conducted enrichment analyses on the CpG probes and on the genes nearest to them. We used eFORGE 2.0[34] to test for enrichment among age or longevity DMPs that either increase or decrease in methylation in comparison with five histone marks and 15 chromatin states mapped in cell lines by the Epigenomics Roadmap Consortium (http://www.ncbi.nlm.nih.gov/

epigenomics). Bat wing tissue is unusual in that it contains epithelial skin, muscle, blood, and elastin[52]. Consequently, we limited enrichment analyses to data from cell lines derived either from skin, blood, or muscle. We also restricted the analysis to probes mapped in a bat genome at least 1 kb apart. We used *Demodus rotundus* to provide a background probe set but obtained very similar results by using other bat genomes, e.g., *Eptesicus fuscus* or *Pteropus vampyrus*, available in eFORGE as backgrounds for the mammalian methylation array. We present enrichment values for each DMP set as the $-\log10 p$ binomial value and consider those outside the 95th percentile of the binomial distribution after correction for multiple testing[64] as significant.

We identified putative transcription factors that could utilize open chromatin and bind to the DNA by testing for enrichment in each DMP set for predicted binding sites among the probes on the mammalian methylation array. Binding sites were included if their FIMO (Find Individual Motif Occurrence) *p*-value was less than 10e−5. FIMO scans were performed using the MEME suite (v.4.12.0, available at http://meme-suite.org/doc/download.html). Bedtools (v.2.25.0) were used to intersect the mammalian methylation array file and provide probe-to-TF motif annotations. We then used a hypergeometric test (phyper) to evaluate overlap between probe sets and transcription factor motifs obtained from four transcription factor databases: TRANSFAC[65], UniPROBE[66], HT-Selex[67], and JASPAR[68]. Redundant transcription factor motifs were then consolidated into clusters[69] to identify distinct transcription factors. The function was inferred using information derived primarily from studies in mice and humans[46].

We used several approaches for determining the type and function of genes associated with age and longevity DMPs. First, we identified the gene (using human orthologs) with the nearest TSS for every mapped probe in each of the four species used to identify longevity DMPs (*R. ferrumequinum*, *Desmodus rotundus*, *Myotis myotis*, and *Molossus molossus*). We then used the subsequent lists of unique genes for each species as background for enrichment tests. While the number of probes near a given gene varies considerably, each gene was covered on average by five probes. The number of unique genes with an identifiable human ortholog near a probe was 4918 in *R. ferrumequinum*, 4693 in *M. molossus*, 4611 in *M. myotis*, and 4534 in *D. rotundus*, reflecting the variation in the number of mapped probes (Supplementary Table 1). Given that the probes were designed to align to regions conserved across all mammals, we suspect some of the differences across species in gene associations reflect variation in genome assembly or annotation. In addition, an important caveat to keep in mind is that the CpGs on the array do not randomly sample the genome[57]. Thus, even when we use mapped probes or the genes near them as background for enrichment tests, there is potential for bias given that the probes are in conserved regions. We assumed a gene was associated with hypermethylated DMPs if it had more hypermethylated than hypomethylated sites nearest its TSS and vice versa. We present results in the text for DMP-gene associations for *M. molossus* because it was the only short-lived species with a genome, but we summarize the DMP-gene associations for the other three species in Supplementary Fig. 5. Because we anticipated the mechanisms responsible for causing increases in methylation over time likely differs from those causing decreases, we conducted separate enrichment tests for genes with hypermethylated and hypomethylated sites associated with age and longevity using Panther v.16[70] in relation to biological process, molecular function, and protein class. We carried out enrichment tests using genes with DMPs in promoter regions because promoter regions showed enrichment for hypermethylated sites. To minimize redundancy due to the hierarchical organization of gene ontologies (GO), we present no more than three significant (after FDR correction) GO terms from each parent–child group. All significant GO terms can be found in the Source Data files. We also used the significant age and longevity gene promoter lists to predict possible transcription factor regulators using BART, Binding Analysis for Regulation of Transcription[47], which correlates the cis-regulatory profile derived from a gene set to the genomic binding profiles of 918 transcription factors using over 7000 human ChIP-seq datasets. We report the Irwin-Hall *P*-value, which indicates the significance of the rank integrated over three test statistics[47].

In addition, we carried out additional analyses to assess gene function using three relevant gene lists. The first utilized a list of 394 genes associated with changes in methylation over the lifespan of dogs[17]. This study assayed over 50,000 CpG sites for 104 known-aged labrador dogs, and included methylation data from mice and humans, to identify 198 hypermethylated and 196 hypomethylated sites, with most of the hypermethylated sites near genes associated with anatomical development. By comparing gene lists, we identified the number of positive (and negative) methylated genes in the dog list that occur in the genome of each bat, and then used the number of genes in the bat, as well as the number of age-related genes in the bat and the number that overlap to calculate the probability associated with the overlap in each methylation direction. We used the R program phyper to conduct a Fisher's Exact Test (FET) using the hypergeometric distribution.

The second test utilized a list of 576 genes that have been documented to mutate frequently in over 10,864 human tumor cases. We downloaded v1.25.1 from the Genome Data Center of the National Cancer Institute (https://portal.gdc.cancer.gov). As with the dog age genes above, we calculated the probability of overlap between the cancer genes found in the genomes of each of four bat species and both the bat age and longevity gene lists using a FET.

The third test involved comparing a list of 4723 innate immunity genes downloaded from https://www.innatedb.com (Aug 14, 2020). As with the cancer gene list, we calculated the probability of overlap between the immunity genes

found in the genome of the four bat genomes and both the bat age and longevity gene lists using a FET.

**Reporting summary**. Further information on research design is available in the Nature Research Reporting Summary linked to this article.

## Data availability

All data used in this study are freely available. Normalized methylation values for each sample, along with sample metadata, are available from NCBI GEO as series GSE164127 (https://www.ncbi.nlm.nih.gov/geo/query/acc.cgi?acc=GSE164127). The design of the Illumina microarray (HorvathMammalMethylChip40) is available from the Gene Expression Omnibus (GEO) at NCBI as platform GPL28271 (https://www.ncbi.nlm.nih.gov/geo/query/acc.cgi?acc=GPL28271). Microarray probe annotations for ten bat genomes are available from the Digital Repository at the University of Maryland (DRUM) at http://hdl.handle.net/1903/26373. Coefficients from the penalized regressions used to estimate bat age for different taxonomic groups are available at https://doi.org/10.6084/m9.figshare.c.5257271. Transcription factor databases used in this study are available as follows: TRANSFAC (http://gene-regulation.com/pub/databases.html), UniPROBE (http://thebrain.bwh.harvard.edu/uniprobe/), HT-Selex (https://ccg.epfl.ch/htpselex/) and JASPAR (http://jaspar.genereg.net/downloads/). Source data for figures are provided with this paper.

## Code availability

R code for implementing the analyses described in this paper is available at https://doi.org/10.6084/m9.figshare.c.5257271.

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

## Acknowledgements

This work was supported by a Paul G. Allen Frontiers Group grant to S.H., the University of Maryland, College of Computer, Mathematical and Natural Sciences to G.S.W., an Irish Research Council Consolidator Laureate Award to E.C.T., a UKRI Future Leaders Fellowship (MR/T021985/1) to S.C.V. and a Discovery Grant from the Natural Sciences and Engineering Research Council (NSERC) of Canada to P.A.F. S.C.V. and P.D. were supported by a Max Planck Research Group awarded to S.C.V. by the Max Planck Gesellschaft, and S.C.V. and E.Z.L. were supported by a Human Frontiers Science Program Grant (RGP0058/2016) awarded to S.C.V. L.J.G. was supported by an NSERC PGS-D scholarship. We thank the Neurogenomics Core at UCLA for laboratory assistance, A. Lollar for providing *Tadarida* samples, M. Brooks for sharing a new maximum recorded lifespan for *Pteropus giganteus*, K. Bennett for graphical assistance, and to the Banbury Center, Cold Spring Harbor Labs for hosting the workshop that inspired this collaboration.

## Author contributions

G.S.W., D.M.A., and S.H. conceived and designed the study. D.M.A., B.D.A., H.C.B., G.G.C., L.N.C., D.K.N.D., P.D., P.A.F., L.J.G., N.J.F., A.V.G., L.G., E.H., G.J., M.K., E.Z.L., F.M., R.A.M., L.G.H.M., J.J.F.M., M.N., B.P., M.L.P., R.D.R., E.C.T., S.C.V., G.S.W., and D.Z. provided or prepared samples. D.M.A., C.E.B., A.H., A.T.L., S.H., C.Z.L., J.A.R., G.S.W., J. Zhang, and J. Zoller analyzed and interpreted data. G.S.W., D.M.A., and S.H. drafted the article. All authors provided comments to improve intellectual content and approve the final version.

## Competing interests

S.H. is a founder of the non-profit Epigenetic Clock Development Foundation which plans to license several patents from his employer UC Regents. These patents list S.H. as inventor. The other authors declare no competing interests.
