## [Peer Review File · Nature Communications]

Reviewer #1 (Remarks to the Author):

The manuscript by Wilkinson et al describes development of methylation aging clocks for 26 species of bats. Bats are the second most species-rich clade after rodents, therefore developing an aging clock that works for a diverse set of bat species is a spectacular achievement. Bats is a very important clade for biological research because of their unique ecology, longevity and immunology. Having this tool that allows estimating bat age with high precision will facilitate studies in a broad range of life sciences. The manuscript also goes further to identify genes and pathways associated with aging and longevity in bats. These genes are enriched for pathways involved in cancer and innate immunity, which is an exciting result. Overall, this is an impressive and well executed study that will be of high interest to wide readership.

I have no major concerns. A minor suggestion is to clarify the definition of aging versus longevity DMPs.

Reviewer #2 (Remarks to the Author):

This is a tour de force study that analyzed patterns of DNA methylation across 26 species of bats, with samples collected in field and laboratory studies, where the age of animals is known. The sheer number of samples analyzed with the Horvath mammal chip provides valuable information for future studies. Bats are the longest-lived mammals based on the lifespan expected based on body mass, and even some of the smallest bats can live more than 40 years in the wild. Getting together this dataset, with the ages of animals up to above 20 years old, is amazing. It is also impressive that the authors were able to develop such an accurate epigenetic clock that represents all bats. While the study is very nice, its presentation may be improved.

It would be useful to begin Results with information on samples analyzed, such as sample number, tissue source, number of species, source of samples, ages, and other characteristics. Then, some basic information would be useful, such as average methylation. And only then the clocks. Otherwise, it is hard to follow the paper by reading through Introduction and then Results. Table 1 lists the samples, but the description is in Methods. Also, the total number of samples is not indicated anywhere.

How many of the 37,500 sites are covered across the bat genomes?

How many CpGs represent the bat clocks and clocks for various bat species? Is there information with weights of the sites that would allow other researchers to repeat the analyses?

Fig. 2: the legend indicates 26 species, but the figure shows only 23 points in both panels.

Maximum lifespan is known to be not do reliable in bats. Was there any attempt to use other life history traits, such as gestation period and time to maturity for the analyzed species?

It would be useful to see figures for selected species in the main text. Consider moving supplementary figure 1 or its subset to the main part.

Fig. 2 shows longevity as longevity quotient. How would the figure look if maximum lifespan is shown, or at least some other life history traits? Another way to show the findings is to show microbats separately and perhaps some other groups of bats too.

Fig. 1B. Many samples where the age is predicted to be higher than the actual age belong to *Rhinolophus ferrumequinum* and *Rhynchonycteris naso*. Is there anything special about these species or their sample manipulation?

Reviewer #3 (Remarks to the Author):

Wilkinson et al., describe methylation profiling of ~700 tissue samples of age-known bats spanning species with a wide range of expected lifespans. As in previous work, they identify methylation sites that can predict chronological age. They further show that rates of change at these sites are

inversely correlated with species longevity. They identify longevity-associated SNPs and show that these are enriched near genes with functional annotations for developmental processes, immunity and cancer.

Bats are an interesting and appropriate choice for this study. The clade includes a (very) large number of species that display a wide range of expected lifespans.

Distinct statistical methods are used to identify predictors of chronological age versus longevity associated methylation sites. The methods described are appropriate and established. I wonder however, why the chronological age prediction did not include a (random effect) term for species. This would seem an obvious approach to develop a species-specific predictor that benefits from partial pooling of information across species. The authors use this approach when predicting longevity associated SNPs.

I find it puzzling that highly conserved (across mammals) probe sequences can be associated with different genes in different species. Are the probe sequences homologous (shared by descent from a common ancestor)? Is the gene-assignment just a matter of degree (slightly closer to gene A vs. gene B)? Or is there larger scale relocation? Please explain.

I am confused about the second paragraph on page 8. Where did the multi-tissue histone methylation data come from. It is not described in the methods. Please clarify and do a bit more framing so we can better understand the reasoning behind this analysis.

The results of this work suggest a transcriptional program that unfolds with age - while it may be speculative, this has some profound implications (ie., is aging a programmed process or simply entropy doing its thing?). It seems likely (to me) that the associated methylation changes are secondary to a causal process driving this program. I want to challenge the authors to make an appropriately qualified statement about causality here. The concluding sentence, which uses the term "facilitates", seems to hint at a causal interpretation in which methylation changes are driving the process. What do the authors think: Is it causal, secondary or chicken-and-egg?

Lastly, and my only major concern, is that the data availability statement is inadequate. Data and software (R scripts) supporting this work should be released on a repository, e.g. Figshare, where researchers can access them without being subject to the authors' interpretation of a "reasonable" request.

Responses to Reviewer Comments (in bold)

We would like to thank all three reviewers for their useful suggestions. Below we describe changes we made or provide rationale for why we decided not to make a change. In addition, since submitting the manuscript, we discovered an error in the annotation pipeline which resulted in the removal of about 5% of the mapped CpG sites for each species. We also included additional DNAm data for 18 known-aged individuals for one species, *Eptesicus fuscus*. After correcting the annotations and incorporating the new data, we completely reanalyzed the data. No patterns or conclusions have changed, but a few relationships have been strengthened. We believe the manuscript is improved as a consequence of these changes.

As requested, all data are or will be publicly available. The methylation data have been uploaded to NCBI GEO as GSE164127. They are currently embargoed, but can be accessed using token "itcpikiypjyxdyj". Once this paper has been accepted, the data will be released to the public. As requested, the clock coefficients and the R scripts used have been uploaded to Figshare and are available at <https://doi.org/10.6084/m9.figshare.c.5257271>. In addition to the supplement, we have included a zipped folder that contains the files used for making the figures as Source Data.

Reviewer #1 (Remarks to the Author):

The manuscript by Wilkinson et al describes development of methylation aging clocks for 26 species of bats. Bats are the second most species-rich clade after rodents, therefore developing an aging clock that works for a diverse set of bat species is a spectacular achievement. Bats is a very important clade for biological research because of their unique ecology, longevity and immunology. Having this tool that allows estimating bat age with high precision will facilitate studies in a broad range of life sciences. The manuscript also goes further to identify genes and pathways associated with aging and longevity in bats. These genes are enriched for pathways involved in cancer and innate immunity, which is an exciting result. Overall, this is an impressive and well executed study that will be of high interest to wide readership.

I have no major concerns. A minor suggestion is to clarify the definition of aging versus longevity DMPs.

We have attempted to clarify the definition of DMPs associated with age or with longevity in the Methods and Results. We also have added a new figure to the supplement that illustrates how DNAm changes with age for the two short-lived species in comparison to the three long-lived species for four significant longevity DMPs. We hope visualizing the patterns in the data will make it easier to understand how we defined longevity DMPs.

Reviewer #2 (Remarks to the Author):

This is a tour de force study that analyzed patterns of DNA methylation across 26 species of bats, with samples collected in field and laboratory studies, where the age of animals is known. The sheer number of samples analyzed with the Horvath mammal chip provides valuable information for future studies. Bats are the longest-lived mammals based on the lifespan expected based on body mass, and even some of the smallest bats can live more than 40 years in the wild. Getting together this dataset, with the ages of animals up to above 20 years old, is amazing. It is also impressive that the authors were able to develop such an accurate epigenetic clock that represents all bats. While the study is very nice, its presentation may be improved.

It would be useful to begin Results with information on samples analyzed, such as sample number, tissue source, number of species, source of samples, ages, and other characteristics. Then, some basic information would be useful, such as average methylation. And only then the clocks. Otherwise, it is hard to follow the paper by reading through Introduction and then Results. Table 1 lists the samples, but the description is in Methods. Also, the total number of samples is not indicated anywhere.

Thank you for this suggestion. We now include a brief summary of the samples at the beginning of the Results to help readers interpret Table 1 and state that we used 712 samples to estimate a clock for all species. Subsequent analyses used fewer species, and therefore, fewer samples. We mention that DNAm estimates for probes not mapped in any bat give comparable estimates to human SNP probes, which can be considered as controls, in contrast to probes that map to at least one bat genome, which have higher average values and are much more variable, as would be expected.

How many of the 37,500 sites are covered across the bat genomes?

We now mention the number of probes mapped in at least one bat genome, 35148, at the beginning of the Results. We also give the number of sites found in each of the nine bat genomes for which we have DNAm data in Supplementary Table 1 along with the version of the genome assembly that was annotated.

How many CpGs represent the bat clocks and clocks for various bat species? Is there information with weights of the sites that would allow other researchers to repeat the analyses?

The number of sites/clock varies depending on the data set used. We now provide the coefficients for each clock in a file that is posted on Figshare, which we cite in the Data Availability statement.

Fig. 2: the legend indicates 26 species, but the figure shows only 23 points in both panels.

We only included species with more than 10 samples in Fig 2, which is why three species are missing. We have updated the legend to make that clear.

Maximum lifespan is known to be not do reliable in bats. Was there any attempt to use other life history traits, such as gestation period and time to maturity for the analyzed species?

We respectfully disagree with the assertion that maximum lifespan is unreliable in bats. The maximum of a sample is the nth order statistic. As sample size increases, the nth order statistic should also increase, but is expected to do so by an ever-decreasing amount. When sample sizes are large, any increase will be very small. Sample sizes for the recorded lifespan of many species of bats are often large, so while maximum lifespans likely will continue to increase, it is unlikely they will increase by much. Moreover, the difference in recorded lifespans between the three long-lived species and the two short-lived species that we used to identify longevity DMPs is 20 years or more, even though they have similar body sizes (20-40 g). The three long-lived species also represent three different phylogenetic lineages.

In contrast, gestation length is not easily determined in many bats because some species can store sperm for up to six months while others can delay implantation. Age of first reproduction shows very little variation in bats, with most species reproducing in year 1 and some species waiting to reproduce until year 2. Litter size also shows little variation, with most species giving birth to one pup/season and a few species giving birth to two. Nevertheless, we ran PGLS analyses on time to maturity and gestation length. In contrast to our results on maximum lifespan or longevity quotient, neither reproductive trait was related to the change in DNAm, so we did not include these analyses in the paper.

It would be useful to see figures for selected species in the main text. Consider moving supplementary figure 1 or its subset to the main part.

We have expanded Fig 1 to include four panels, and now include the LOO clock for two species with 40 or more samples to illustrate that clocks can sometimes, but not always, improve if the data are restricted to a single species. We have left the remaining figures in the Supplement for

readers who want to see more examples. As noted below, we also now provide the coefficients for the clocks for each dataset in a file available from Figshare.

Fig. 2 shows longevity as longevity quotient. How would the figure look if maximum lifespan is shown, or at least some other life history traits? Another way to show the findings is to show microbats separately and perhaps some other groups of bats too.

As we show below, the figure for maximum lifespan, adjusted for body size, looks very similar to the figure for longevity quotient. We present the statistical results for this analysis in Supplementary Table 2. We chose to present longevity quotient in Fig 2 because it is widely used in the comparative aging literature and provides a more intuitive measurement than “Residual longevity”. We did not include the figure below because the number of supplementary figures is limited. As we note above, there is no relationship between mean methylation rate and any other life history trait, so adding those plots would not be appropriate. With regard to highlighting groups of bats, we want to point out that the 26 species represent six different families from the two suborders of bats. We did not distinguish megabats and microbats because they are not monophyletic groups. Rhinolophidae and Pteropodidae are in the same suborder, Yinpterochiroptera, i.e. species 4 and 22, which have very different lifespans and methylation rates. This is true for other groups, too. Species 5 and 8 are in the family Phyllostomidae, species 10 and 12 are in Vespertilionidae, and species 9 and 26 are in Molossidae. The fact that the pattern occurs in these different families is why the phylogenetic analysis was significant. We could have color-coded points by family, but we decided instead to highlight the species used to estimate longevity. However, for readers unfamiliar with bat families, we now include the family for each species in Table 1.

Fig. 1B. Many samples where the age is predicted to be higher than the actual age belong to *Rhinolophus ferrumequinum* and *Rhynchonycteris naso*. Is there anything special about these species or their sample manipulation?

There is little, if anything, in common between the *Rhinolophus* and *Rhynchonycteris* samples. The wing punches were taken from wild bats in England or in Costa Rica, respectively. DNA was then extracted by different people in different labs. Each of those labs also processed samples from other species (*Myotis myotis* and *Saccopteryx bilineata*, respectively). DNA methylation from all samples was assayed at UCLA with samples distributed across multiple plates. *Rhynchonycteris* is in the family Emballonuridae, as is *Saccopteryx bilineata* (species 25), so the variation is independent of the phylogeny.

That being said, after reanalysis of the reannotated data, which removed probes that failed to map correctly to each bat genome, the Rhynchonycteris samples no longer are outliers, so presumably incorrect probe annotation contributed to their location in the LOSO analysis.

Reviewer #3 (Remarks to the Author):

Wilkinson et al., describe methylation profiling of ~700 tissue samples of age-known bats spanning species with a wide range of expected lifespans. As in previous work, they identify methylation sites that can predict chronological age. They further show that rates of change at these sites are inversely correlated with species longevity. They identify longevity-associated SNPs and show that these are enriched near genes with functional annotations for developmental processes, immunity and cancer.

Bats are an interesting and appropriate choice for this study. The clade includes a (very) large number of species that display a wide range of expected lifespans.

Distinct statistical methods are used to identify predictors of chronological age versus longevity associated methylation sites. The methods described are appropriate and established. I wonder however, why the chronological age prediction did not include a (random effect) term for species. This would seem an obvious approach to develop a species-specific predictor that benefits from partial pooling of information across species. The authors use this approach when predicting longevity associated SNPs.

We did not include a random effect for species in the elastic-net regressions for age because we wanted to illustrate how well the clock could predict age for samples from each of the species in the dataset or from an unknown species. It is perhaps worth noting that the elastic-net regression using all of the data (i.e. not performing a leave-one-out analysis) fits almost perfectly. Including species as a random effect would have very little influence on model fit.

I find it puzzling that highly conserved (across mammals) probe sequences can be associated with different genes in different species. Are the probe sequences homologous (shared by descent from a common ancestor)? Is the gene-assignment just a matter of degree (slightly closer to gene A vs. gene B)? Or is there larger scale relocation? Please explain.

Figure 4 in the Supplement provides information to address this puzzle. In cases where a probe maps to a different gene, it is often because the probe mapped to an intergenic region and the nearest tss differs between species. Bats have much smaller genomes than humans (typically < 2 Gbp instead of 3.2 Gbp), so intergenic regions are often smaller. They also have different transposable elements and are one of the few mammals with active DNA transposons. A TE insertion in an intron or intergenic region could easily alter the nearest tss for a given probe. In addition, we suspect some differences could be due to assembly quality. Not all of the bat genomes represent chromosomal assemblies. Consequently, a probe could map to a different tss if it is near the end of a scaffold in a species, rather than in the middle of a chromosome. Nevertheless, as Sup. Fig 4c and Fig 4d show, most probes are near the same genes, especially if they are in promoter regions (see also Fig 3c), for the species we used to identify longevity DMPs.

I am confused about the second paragraph on page 8. Where did the multi-tissue histone methylation data come from. It is not described in the methods. Please clarify and do a bit more framing so we can better understand the reasoning behind this analysis.

The multi-tissue histone marks are consolidated from the Epigenomics Roadmap Project (<http://www.roadmapepigenomics.org>). This project has generated several thousand datasets derived from over 100 cell lines from various tissues. Each dataset contains the locations of core histone marks (H3K4me3, H3K4me1, H3K27me3, H3K9me3, and H3K36me3) and predicted chromatin states for each cell type. The eFORGE program (<https://eforge-dev.altiusinstitute.org>) performs randomization tests using probes that map to a specific animal genome and then identifies probes that are nonrandomly associated with each histone mark (or predicted chromatin state). Method details can be found in Breeze, C.E. et al. 2016 Cell Reports. We have added two

additional sentences to the paragraph in question to help reduce confusion.

The results of this work suggest a transcriptional program that unfolds with age - while it may be speculative, this has some profound implications (ie., is aging a programmed process or simply entropy doing its thing?). It seems likely (to me) that the associated methylation changes are secondary to a causal process driving this program. I want to challenge the authors to make an appropriately qualified statement about causality here. The concluding sentence, which uses the term "facilitates", seems to hint at a causal interpretation in which methylation changes are driving the process. What do the authors think: Is it causal, secondary or chicken-and-egg?

We think sites that undergo hypomethylation with age do so largely at random. In contrast, sites that undergo hypermethylation with age are highly nonrandom, and as has been noted before, are near genes associated with development. So yes, we believe there are predictable methylation changes with age. We have added a sentence to the Discussion to indicate that the similarity in hypermethylated age genes between humans, mice, dogs and bats must be the result of shared ancestry. However, because we did not measure any aging phenotypes, we cannot conclude that methylation is driving "aging."

Lastly, and my only major concern, is that the data availability statement is inadequate. Data and software (R scripts) supporting this work should be released on a repository, e.g. Figshare, where researchers can access them without being subject the authors' interpretation of a "reasonable" request.

The methylation data have been uploaded to NCBI GEO as GSE164127. They are currently embargoed, but can be accessed using token "itcpikiypjyxdyj". Once this paper has been accepted, we will release the data to the public. As requested, the clock coefficients and the R scripts used have been uploaded to Figshare and are available at <https://doi.org/10.6084/m9.figshare.c.5257271>. In addition to the supplement, we have included a zipped folder that contains the files used for making the figures as Source Data.

Reviewer #1 (Remarks to the Author):

The authors fully addressed my concerns. I wholeheartedly recommend publishing this important paper.

Reviewer #2 (Remarks to the Author):

Authors addressed all my comments, and the manuscript has improved. This is a very nice study, which I suggest to accept.

Reviewer #3 (Remarks to the Author):

The authors have adequately addressed all concerns and have clarify (in text) some points of confusion.